# An Efficient Approach to Face Emotion Recognition with Convolutional Neural Networks

Christian Białek [1], Andrzej Matiolański [1,2,*] and Michał Grega [1,2]

1 Institute of Telecommunications, AGH University of Science and Technology, 30-059 Kraków, Poland
2 Aiseemo Sp. z o.o., 30-054 Kraków, Poland
* Correspondence: andrzej.matiolanski@agh.edu.pl

**Abstract:** Solutions for emotion recognition are becoming more popular every year, especially with the growth of computer vision. In this paper, classification of emotions is conducted based on images processed with convolutional neural networks (CNNs). Several models are proposed, both custom and transfer learning types. Furthermore, combinations of them as ensembles, alongside various methods of dataset modification, are presented. In the beginning, the models were tested on the original FER2013 dataset. Then, dataset filtering and augmentation were introduced, and the models were retrained accordingly. Two methods of emotion classification were examined: a multi-class classification, and a binary classification. In the former approach, the model returns the probability for each class. In the latter, separate models for each single class are prepared, together with an adequate dataset based on FER2013. Each model recognizes a single emotion from the others. The obtained results and a comparison of the applied methods across different models is presented and discussed.

**Keywords:** facial emotion recognition; image analysis; convolutional neural network; FER2013

## 1. Introduction

Facial expressions contain a lot of valuable and accessible information. The most common purpose of the analysis of facial expressions is to determine the current emotion of a particular person. Many companies (for instance, Google [1] and Microsoft [2]) have already incorporated such solutions into their applications. The problem seems fairly easy when the emotion in the image is obvious. For example, the "happy" emotion is usually simple to identify. However, a difficulty arises when the intensity of the emotional expression is low, and thus predictions based on micro-expressions can be close in their probability values and easily mistaken.Such a situation often occurs with "neutral" and "sad" emotions, when even the slightest changes can make a difference. As we know from experience, sometimes even humans have trouble recognizing the real emotion. The provided solutions can be successfully used as provided or incorporated into more complex solutions; for instance, in the fast-growing field of remote communication with webcams enabled, such as teleconferences or job interviews, suggesting how the other party might feel or react in certain situations.

Artificial intelligence, especially in the field of computer vision, allows for advanced image analysis and inference of hidden features. This makes this technique effective and successful when applied in the problem of object classification in various fields. One such area is emotion recognition based on the facial expression presented in a provided image. In particular, convolutional neural networks (CNNs) have proven to be effective in solving such tasks. First, the layers responsible for feature extraction [3] aim to recognize specific patterns (e.g., micro-expressions) in images. Then the classification layers, based on the input data from previous layers, perform classification (e.g., emotion recognition). The use of CNNs in facial expression recognition (FER) has been the topic of many articles, presenting a wide range of different approaches to the problem. The most popular include transfer learning, custom CNNs, and ensemble models [4].

The dataset used to conduct this research was the well-known FER2013 created by Pierre Luc Carrier and Aaron Courville. It was first introduced in [5] for a Kaggle competition. The dataset consists of 35,887 grayscale images, which are cropped and each of the size of 48 × 48 pixels. The face in the images can be classified into one of seven possible categories. The FER2013 dataset is highly imbalanced, since the number of images per class varies a lot (the most numerous,"happy", contains 8989 images, and the least numerous,"disgust", only 547). As described in [5], the human accuracy on this dataset is about 65 ± 5%. Moreover, some images may be labeled incorrectly, due to the method of their collection, as well as the use of crowd-sourcing for labeling [6]. These dataset characteristics have room for further improvement, which will be discussed in more detail in Section 3. There are other popular datasets available, such as AffectNet [7], JAFFE [8], CK+ [9], and KDEF [10], representing either images taken "in the wild" (achieved accuracy reported in literature is around 75%) or in laboratory conditions (reported accuracy exceeding 90%). However, these datasets are not covered by the scope of this paper.

To conclude the introduction, the unique nature of this manuscript and its distinct contributions to the existing knowledge on facial emotion recognition (FER) are emphasized. Although numerous studies investigating FER methodologies using convolutional neural networks (CNN), including VGG and ResNet architectures, have been conducted, several unique elements are offered by this research that distinguish it from the current state of the art:

- A comprehensive comparative analysis of the existing FER methods that utilize a CNN architecture is presented. Through the side-by-side examination of these methodologies, an in-depth understanding of their respective strengths and weaknesses is offered, highlighting areas where improvements could be made and potential research opportunities may be found. This study goes beyond mere implementation of existing architectures, aiming to enhance the collective understanding of their performance in various scenarios;
- The introduction of two novel models for FER is a key feature of this work. The first model, which prioritizes efficiency, has been designed to achieve commendable performance in a shorter time frame than the previously published models. The second model, an ensemble approach, leverages fewer models than commonly reported in the literature yet outperforms many more complex systems. These innovative models contribute to the continual advancement of FER methodologies, setting new performance and efficiency standards;
- In terms of resources for the research community, significant strides have been made through the meticulous updating of the FER database and making an enhanced version publicly available. This allows researchers to conduct more accurate and relevant analyses using the most current data, fostering progress in the field;
- A divergence from the norm is proposed by the use of binary models in FER—an approach not frequently adopted in existing literature. This addition extends the range of methods available for FER and paves the way for new lines of inquiry and exploration in emotion recognition research.

The rest of this paper is structured as follows: Section 2 describes an overview of works related to this paper. Section 3 explains the FER2013 dataset in depth, along with the applied modification methods. Section 4 presents the selected models. Section 5 compares the obtained results between models presented in this paper, as well as the results obtained by other researchers. Section 6 provides a discussion of the presented approach. Section 7 briefly concludes the paper.

## 2. Related Work

A lot of research effort has been put into the topic of facial emotion recognition. In this analysis, we will focus on the most recent works, while also presenting the most significant efforts from previous years. Several approaches to research can be identified. Some of the researchers focused on enhancing a dataset, either by augmentation or by using external data, in order to improve the performance of the networks. Others focused

on hyperparameter tuning. Another approach was to modify or extend existing model architectures. Finally, much effort, with excellent results, has been applied to creating ensemble models. Luckily, most researchers used the FER2013 dataset, which allows for a quantitative comparison of results.

One of the first works using the FER2013 dataset was that presented in 2013 by Goodfellow et al. [5], which determined the accuracy of the "null model", obtaining 65.5% as an ensemble of a few convolutional neural networks and with use of a TPE hyperparameter optimization method. In the following year, Nguyen et al. [11] focused on the influence of low-level, mid-level, and high-level features on the performance of facial emotion classification. As a result, accuracies of 73.03% and 74.09% were achieved with the use of a single MLCNN and ensemble MLCNNs (multi-level convolutional neural networks), respectively. As a backbone, a single VGGNet-inspired [12] 18-layer network was used.

Pioneering work in using additional features such facial landmarks and utilizing external datasets was presented in 2015 by Zhang et al. [13]. The authors were able to obtain an accuracy of 75.1%. In 2016, Pramerdorfer et al. [14] focused on finding bottlenecks in the existing CNNs, and the removal of one of them was proven to increase classification performance on the FER2013 dataset. An ensemble of 8 such networks achieved an accuracy of 75.2%. The same year, Kim et al. [15] introduced a two-level hierarchical committee consisting of a number of deep CNNs. By ensembling 36 networks and averaging outputs, an accuracy of 72.72% was achieved. A year later, Connie et al. [16] proposed a hybrid CNN–SIFT (scale invariant feature transform) aggregator, which was a combination of three models: a single CNN, a CNN with SIFT, and a CNN with dense SIFT. It obtained an accuracy of 73.4%, albeit lower than the previous efforts.

In 2018, Jun et al. [17] proposed a 19-layer VGGNet-inspired model with slight modifications, such as adding dropout to a fully connected layer. What is more, before mirroring the training, images were randomly cut down by 4 pixels reducing their resolution to 44 × 44 pixels. The highest accuracy achieved by the best configuration was 73.06%. In 2019, Hua et al. [18] presented an ensemble of 3 models (the best individual model obtained an accuracy of 68.18%), with an increasing number of convolutional layers and achieved an accuracy of 71.91%. This result was not a significant improvement in accuracy over the previous models, but again justified use of the ensemble approach. In the same year, Porușniuc et al. [19] achieved an accuracy of 71.25% by using the RESNET50 model with pretrained weights trained on the VGGFace2 [20] dataset. Besides a basic data augmentation, they also performed contrast limited adaptive histogram equalization (CLAHE), which resulted in more emphasized contours and a higher contrast of the images. Also in 2019, Georgescu et al. [21] proposed a complicated model with an accuracy of 75.42%. Additionally to the CNNs, the BOVW (bag of visual words) model was used, as well as linear SVM and local learning techniques.

The following years have shown an increase in the interest in this topic, which can be linked to general interest in machine learning methods and their overall improvement and development. In 2020, Kusuma et al. [22] presented a standalone model based on the VGG-16 and achieved an accuracy of 69.40%. The model was modified in order to benefit from global average pooling (GAP) as its last pooling layer. The experiments took into account many factors, such as the use of different optimizers, such as SGD, Adam, SWATS (switching from Adam to SGD), balanced and imbalanced data distributions, layer freezing, and the use of batch normalization. In addition, Riaz et al. [23] proposed a shallow eXnet network with only 4.57 million parameters, along with some up-to-date data augmentation methods, such as cutout, mixup, and the combination of these two, which achieved an accuracy of 73.54%. Jia et al. [24] obtained an accuracy of 71.27% using an ensemble of 3 models (inspired by ALexNet, VGGNet, and ResNet) with a support vector machine (SVM) as a classifier. Khanzada et al. [25] achieved an accuracy of 74.8% on the original FER2013, and 75.8% with auxiliary data as an ensemble of 7 models. Pham et al. [26] proposed a new residual masking network consisting of four residual masking blocks. It

achieved an accuracy of 74.14% and 76.82% using single-model and ensemble method (as an ensemble of 6 CNNs) approaches, respectively. The authors utilized auxiliary training data in their work.

In 2021, Minaee et al. [27] proposed an attentional convolutional network and achieved an accuracy of 70.02% with the use of a spatial transformer (affine transformation was applied), whose main task was to concentrate on a specific part of the image. Khaireddin et al. [28] proposed a single network based on VGGNet. By experimenting with various optimizers and learning rate schedulers, they managed to obtain an accuracy of 73.28%. Another research work proposing hyperparameter optimization was conducted by Vulpe-Grigoraşi et al. [29], which utilized a random search algorithm to find the model with the best performance. It achieved an accuracy of 72.16%.

The most recent (2022) efforts were those of Pecoraro et al. [30], who achieved a 74.42% accuracy using a single model by applying a local multi-head channel (LHC) self-attention module and a ResNet34v2 network as a backbone. In addition, Fard et al. [31] presented adaptive correlation (Ad-Corre) loss with Xception as a backbone network and obtained an accuracy of 72.03%. Furthermore, the RESNET50 model was also examined, achieving an accuracy of 68.25% without Ad-Corre and 71.48% with Ad-Corre (which gave an +3.23% accuracy boost). Another research paper that is worth mentioning is the one presented by Akhand et al. [32], which presented an excellent overview of the use of transfer learning models in this problem. This research, however, cannot be compared to the others in terms of accuracy, as it did not make use of the FER2013 dataset.

A summary of the results of the research presented above is given in Section 5.4. We took the findings of the aforementioned related works into consideration, especially the added value of ensemble methods, a careful approach, and use of the dataset. We started with various modifications of the dataset and then proposed networks with only a few layers, as well as transfer learning ones. In order to boost the performance even higher a method of model ensembling was used in our research. Besides the accuracy, supplementary metrics were introduced, such as precision, recall, and F1-score. In addition to multi-class classification, a binary classification approach is also proposed in the following sections.

### 3. Database and Augmentation

As stated in the introduction, FER2013 [5] was used as the dataset to conduct this research. It was randomly divided into 3 sets with the given split ratio: 80% training, 10% validation, and 10% test sets. More or less every grayscale image in the dataset shows a cropped face of the size 48 × 48 pixels.

First, the prepared models were tested on the original dataset. Then, the manually filtered FER2013 was introduced, where images not presenting a human face were removed and those clearly mislabeled were subjectively reassigned to a different, matching class. Finally, all classes were balanced using basic methods of data augmentation, such as zooming, rotating, and mirroring.

When it came to binary classification, a separate set based on the filtered FER2013 was prepared per class, containing only two possible labels—*chosen emotion* and *other*. Training sets for the binary approach were augmented in the same way as previously, in order to maintain a balance between the two classes. Below, the prepared datasets are described in more detail, with the number of training samples in each dataset presented in Table 1.

**Table 1.** Number of training samples per class in particular datasets.

|  | Original FER2013 | Balanced Original FER2013 | Filtered FER2013 | Balanced Filtered FER2013 |
|---|---|---|---|---|
| Angry | 3995 | 8000 | 3623 | 8000 |
| Disgust | 436 | 8000 | 442 | 8000 |
| Fear | 4097 | 8000 | 3711 | 8000 |
| Happy | 7215 | 8000 | 6816 | 8000 |
| Neutral | 4965 | 8000 | 4493 | 8000 |
| Sad | 4830 | 8000 | 5132 | 8000 |
| Surprise | 3171 | 8000 | 3093 | 8000 |

### 3.1. Original FER2013

The original FER2013 contains 35,887 grayscale images randomly split into 3 sets: training—28,706, validation—3585, and test—3596. More or less every image shows only one cropped face of a size of 48 × 48 pixels and is labeled as one of 7 emotions: "angry", "disgust", "fear", "happy", "neutral", "sad", or "surprise". Exemplary images are presented in Figure 1. The dataset itself is imbalanced, as the sizes of the classes vary greatly. It is worth mentioning that the FER2013 presents a wide range of images taken in the wild, differing in the position of the face, brightness, and distance from the camera. The people depicted in the images also vary in age, race, and gender. Moreover, the actors express emotions with different intensities.

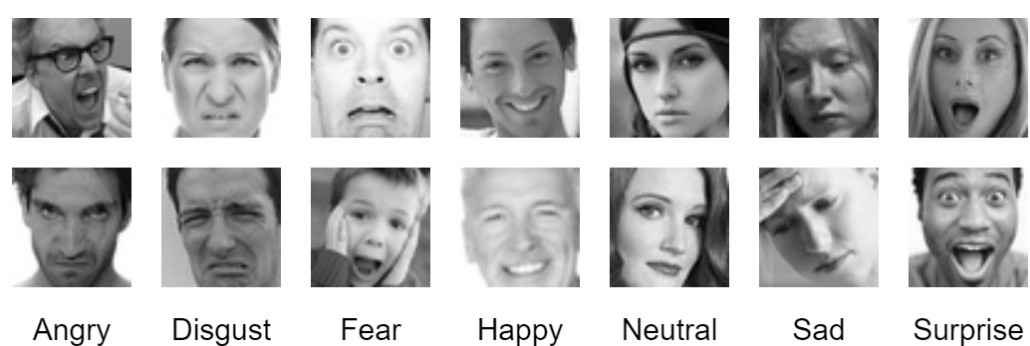

    Angry     Disgust     Fear     Happy     Neutral     Sad     Surprise

**Figure 1.** Selected examples of the original FER2013 dataset.

All of these characteristics allow humans to achieve about a 65 ± 5% accuracy on this dataset, possibly having a high Bayes rate [5]. In the same paper, the "null model" was described as an ensemble [4] of CNNs, which gave a result of a 65.5% accuracy.

### 3.2. Original Balanced FER2013

The original, balanced FER2013 is an augmented version of the original dataset. To balance the differences in the number of images between classes, basic methods of augmentation were used:

- Rotation. Rotating images randomly in the range between 10° clockwise and 10° counterclockwise;
- Mirror. Mirroring images only horizontally, as vertical mirroring does not suit the characteristics of the dataset;
- Zoom. Zooming images with a minimum zoom factor of 1.1 and a maximum of 1.2 (a random value between 1.1 and 1.2 was chosen).

The augmentation process resulted in a balanced dataset of 8000 training samples per class. The test and validation sets were the same as in the original FER2013 and were not augmented.

### 3.3. Filtered FER2013

In the filtered FER2013, the dataset was manually cleaned, in order to remove non-face images (presented in Figure 2), or those which were clearly not corresponding to any particular class. In the case of explicit mislabeling, the picture was subjectively relabeled to the proper category. The resulting dataset consisted of 34,140 images split into 3 sets: training—27,310, validation—3410, and test—3420. Figure 3 shows images that can be classified as different emotions, but as stated before, such a decision is very subjective.

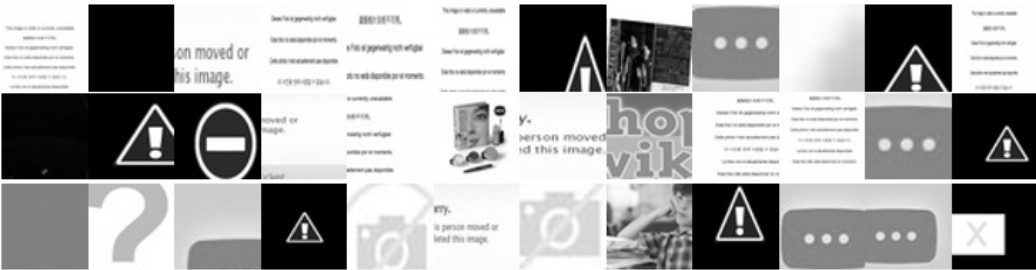

**Figure 2.** Non-face images.

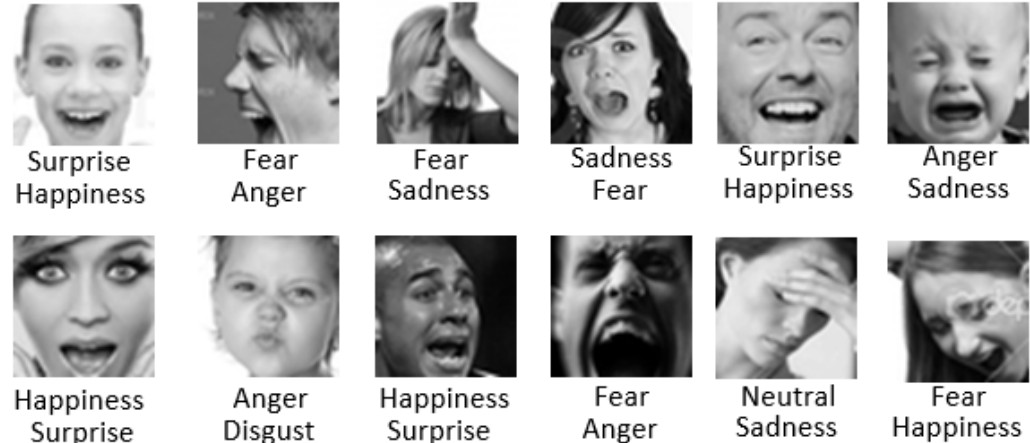

**Figure 3.** Images with ambiguous labels [6].

### 3.4. Filtered Balanced FER2013

The filtered and balanced FER2013 was based on the filtered dataset described above. Its training set was augmented in the same manner as the original, balanced FER2013. This resulted in 8000 training samples per class. The test and validation sets were the same as in the filtered FER2013 and were not augmented.

### 3.5. Binary Datasets

Binary datasets were prepared for every individual emotion. This resulted in assembling seven separate datasets, each having only two possible labels—*chosen class* and *other*.

Initially, from the training samples of the filtered FER2013, random 1000 images per class were taken. The "disgust" emotion did not provide enough samples, so it was taken in full.

Datasets were constructed in such a way that the *other* class constituted of 1000 images of each emotion, except the emotion that the particular binary classification model aimed to identify.

In total, the *other* class was composed of 6000 samples. The *chosen emotion* class consisted of the corresponding class of the filtered FER2013, augmented up to 6000 images in the same manner as previously.

Only the "happy" set was randomly undersampled to meet the criteria, and the remaining images were moved to the validation set. The test set was the same as in the balanced and filtered FER2013.

All classes, except the chosen one, were collected together and then randomly undersampled, in order to keep the balance with the emotion being classified. Table 2 presents the size of the training, validation, and test sets per class for every binary dataset.

**Table 2.** Number of training, validation, and test samples per class.

|  | Training | Validation | Test |
|---|---|---|---|
| Angry/Other | 6000 | 452 | 454 |
| Disgust/Other | 6000 | 55 | 56 |
| Fear/Other | 6000 | 463 | 465 |
| Happy/Other | 6000 | 1667 | 852 |
| Neutral/Other | 6000 | 561 | 563 |
| Sad/Other | 6000 | 640 | 642 |
| Surprise/Other | 6000 | 772 | 388 |

## 4. Proposed Models

Research was carried out on four models in total. Two of them were custom CNN models, and the other two were developed using the transfer learning technique. A convolutional neural network (CNN) was used for its capabilities of feature extraction as performed by the convolutional layers. Then, the classification layers were responsible for emotion recognition [33]. The generic architecture of a such CNN is presented in Figure 4.

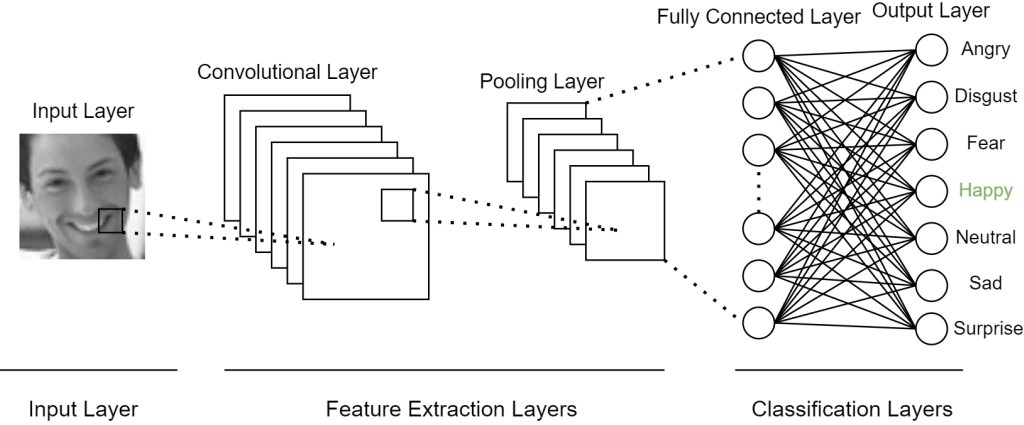

**Figure 4.** Basic architecture of a convolutional neural network for emotion recognition.

Below, the architectures of the applied CNN models are presented, along with the applied parameters and hyperparameters.

### 4.1. Five-Layer Model

The five-layer model was a standard CNN network with five convolutional layers, followed by batch normalization, which decreased the internal covariate shift [34]. This resulted in a faster training time and a higher accuracy [35]. Then, max pooling was used, taking the maximum value of each region of the pooling [36]. To reduce the overfitting, a dropout was applied as a regularizer, making the model more generalized [37].

In the case of the five-layer CNN, a fully-connected layer was implemented. The architecture of the five-layer model is presented in Table 3. Table 4 shows the parameters and hyperparameters used for the five-layer model. Similar ones were used in the six-layer model discussed below, except for the lack of dropouts in the fully connected layers.

**Table 3.** The five-layer CNN model architecture.

| Layer (Type) | Output Shape | Param # |
|---|---|---|
| Conv2D | (None, 48, 48, 128) | 1280 |
| BatchNormalization | (None, 48, 48, 128) | 512 |
| MaxPooling2D | (None, 24, 24, 128) | 0 |
| Dropout | (None, 24, 24, 128) | 0 |
| Conv2D | (None, 24, 24, 256) | 295,168 |
| BatchNormalization | (None, 24, 24, 256) | 1024 |
| MaxPooling2D | (None, 12, 12, 256) | 0 |
| Dropout | (None, 12, 12, 256) | 0 |
| Conv2D | (None, 12, 12, 512) | 1,180,160 |
| BatchNormalization | (None, 12, 12, 512) | 2048 |
| MaxPooling2D | (None, 6, 6, 512) | 0 |
| Dropout | (None, 6, 6, 512) | 0 |
| Conv2D | (None, 6, 6, 1024) | 4,719,616 |
| BatchNormalization | (None, 6, 6, 1024) | 4096 |
| MaxPooling2D | (None, 3, 3, 1024) | 0 |
| Dropout | (None, 3, 3, 1024) | 0 |
| Conv2D | (None, 3, 3, 1024) | 9,438,208 |
| BatchNormalization | (None, 3, 3, 1024) | 4096 |
| MaxPooling2D | (None, 1, 1, 1024) | 0 |
| Dropout | (None, 1, 1, 1024) | 0 |
| Flatten | (None, 1024) | 0 |
| Dense | (None, 512) | 524,800 |
| BatchNormalization | (None, 512) | 2048 |
| Dropout | (None, 1024) | 0 |
| Dense | (None, 256) | 131,328 |
| BatchNormalization | (None, 256) | 1024 |
| Dropout | (None, 256) | 0 |
| Dense | (None, 7) | 1799 |
| **Total params:** | | **16,307,207** |
| **Trainable params:** | | **16,299,783** |
| **Non-trainable params:** | | **7424** |

**Table 4.** Applied parameters and hyperparameters for the five-layer model.

| Parameter/Hyperparameters | Value |
|---|---|
| Image size | $48 \times 48 \times 1$ |
| Batch size | 32 |
| Epochs | 100 |
| Kernel size | $3 \times 3$ |
| Max pooling | $2 \times 2$ |
| Activation function | ReLU (Rectified Linear Unit) |
| Dropout | 25% (in CONV layers) & 50% (in FCL of 5 layer model) |
| Adam optimizer | 0.001 |
| Loss function | Categorical Cross-Entropy |
| Output layer activation function | Softmax |

*4.2. Six-Layer Model*

The six-layer CNN model was composed in a similar manner as the previous model, with the exception of an additional layer, as well as the use of global average pooling (GAP) instead of fully connected layers (FCL). The architecture of the six-layer model is presented in Table 5. Table 6 shows the parameters and hyperparameters used for the six-layer model.

**Table 5.** The six-layer CNN model architecture.

| Layer (type) | Output Shape | Param # |
|---|---|---|
| Conv2D | (None, 48, 48, 128) | 1280 |
| BatchNormalization | (None, 48, 48, 128) | 512 |
| MaxPooling2D | (None, 24, 24, 128) | 0 |
| Dropout | (None, 24, 24, 128) | 0 |
| Conv2D | (None, 24, 24, 256) | 295,168 |
| BatchNormalization | (None, 24, 24, 256) | 1024 |
| MaxPooling2D | (None, 12, 12, 256) | 0 |
| Dropout | (None, 12, 12, 256) | 0 |
| Conv2D | (None, 12, 12, 512) | 1,180,160 |
| BatchNormalization | (None, 12, 12, 512) | 2048 |
| MaxPooling2D | (None, 6, 6, 512) | 0 |
| Dropout | (None, 6, 6, 512) | 0 |
| Conv2D | (None, 6, 6, 1024) | 4,719,616 |
| BatchNormalization | (None, 6, 6, 1024) | 4096 |
| MaxPooling2D | (None, 3, 3, 1024) | 0 |
| Dropout | (None, 3, 3, 1024) | 0 |
| Conv2D | (None, 3, 3, 1024) | 9,438,208 |
| BatchNormalization | (None, 3, 3, 1024) | 4096 |
| MaxPooling2D | (None, 1, 1, 1024) | 0 |
| Dropout | (None, 1, 1, 1024) | 0 |
| Conv2D | (None, 1, 1, 2048) | 18,876,416 |
| BatchNormalization | (None, 1, 1, 2048) | 8192 |
| MaxPooling2D | (None, 1, 1, 2048) | 0 |
| Dropout | (None, 1, 1, 2048) | 0 |
| GlobalAveragePooling2D | (None, 2048) | 0 |
| Dense | (None, 7) | 14,343 |
| **Total params:** | | **34,545,159** |
| **Trainable params:** | | **34,535,175** |
| **Non-trainable params:** | | **9984** |

**Table 6.** Applied parameters and hyperparameters for the six-layer model.

| Parameter/Hyperparameters | Value |
|---|---|
| Image size | $48 \times 48 \times 1$ |
| Batch size | 32 |
| Epochs | 100 |
| Kernel size | $3 \times 3$ |
| Max pooling | $2 \times 2$ |
| Activation function | ReLU (Rectified Linear Unit) |
| Dropout | 25% |
| Adam optimizer | 0.001 |
| Loss function | Categorical Cross-Entropy |
| Output layer activation function | Softmax |

### 4.3. Transfer Learning Models

Transfer learning (TL) is a method of using already pretrained models in the same or a similar task. This can improve the performance, as well as save time and resources while training, as the resulting new model utilizes the knowledge previously learned [38,39]. The pretrained models were based on the VGGFace [40] and VGGFace2 [20] datasets (containing 2.6 and 3.31 million images, respectively), which refer to the related topic of facial recognition. For both models, only the convolutional layers were preserved, as they are responsible for feature extraction and are capable of transferring the learned patterns to the new problem, making the classification layers more efficient. Due to the input specification of these models, the images were resized and channel replication was performed accordingly. Their architectures are presented in Tables 7 and 8.

**Table 7.** A simplified pretrained VGG16 model architecture without the four last convolutional layers.

| Layer (Type) | Output Shape | Param # |
|---|---|---|
| *VGG16 base model* | | |
| Conv2D | (None, 14, 14, 128) | 589,952 |
| BatchNormalization | (None, 14, 14, 128) | 512 |
| MaxPooling2D | (None, 7, 7, 128) | 0 |
| Dropout | (None, 7, 7, 128) | 0 |
| Conv2D | (None, 7, 7, 256) | 295,168 |
| BatchNormalization | (None, 7, 7, 256) | 1024 |
| MaxPooling2D | (None, 3, 3, 256) | 0 |
| Dropout | (None, 3, 3, 256) | 0 |
| Conv2D | (None, 3, 3, 512) | 1,180,160 |
| BatchNormalization | (None, 3, 3, 512) | 2048 |
| MaxPooling2D | (None, 1, 1, 512) | 0 |
| Dropout | (None, 1, 1, 512) | 0 |
| Conv2D | (None, 1, 1, 1024) | 4,719,616 |
| BatchNormalization | (None, 1, 1, 1024) | 4096 |
| MaxPooling2D | (None, 1, 1, 1024) | 0 |
| Dropout | (None, 1, 1, 1024) | 0 |
| GlobalAveragePooling2D | (None, 1024) | 0 |
| Dense | (None, 7) | 7175 |
| **Total params:** | | **16,794,823** |
| **Trainable params:** | | **6,795,911** |
| **Non-trainable params:** | | **9,998,912** |

**Table 8.** A simplified pretrained RESNET50 model architecture.

| Layer (Type) | Output Shape | Param # |
|---|---|---|
| *RESNET50 base model* | | |
| Flatten | (None, 2048) | 0 |
| Dense | (None, 2048) | 4,196,352 |
| Dense | (None, 7) | 14,343 |
| **Total params:** | | **27,771,847** |
| **Trainable params:** | | **27,718,727** |
| **Non-trainable params:** | | **53,120** |

The input images had a size of 224 × 224 × 3. Both models were trained on 100 epochs, with a batch size of 32. As in the five-layer and six-layer models, the network was characterized by

- Adam as the optimizer, with a learning rate of 0.0001;
- Categorical cross-entropy as the loss function;
- Softmax as the output layer activation function.

In the case of added parts of the networks, the kernel size, max pooling, activation function, and dropout were not changed and were the same as in Tables 4 and 6.

### 4.4. Binary Models

Binary models tend to achieve better performance in individual classes than multi-class models. When the quantity of possible labels is reasonable, binary models can be used in multi-class classification problems. However, with the increase of the number of classes the inference time becomes slower and the resource utilization becomes higher [41]. The combination of several binary models is called one-vs-rest (OVR) or one-vs-all (OVA) [42]. Its simplified architecture is presented in Figure 5.

The proposed binary models were the same as those used in the multi-class classification but correspondingly converted to identify only a particular class or those representing the rest. The last dense layer of each binary model had the value of (None, 1). Parameters and hyperparameters were the same as before except

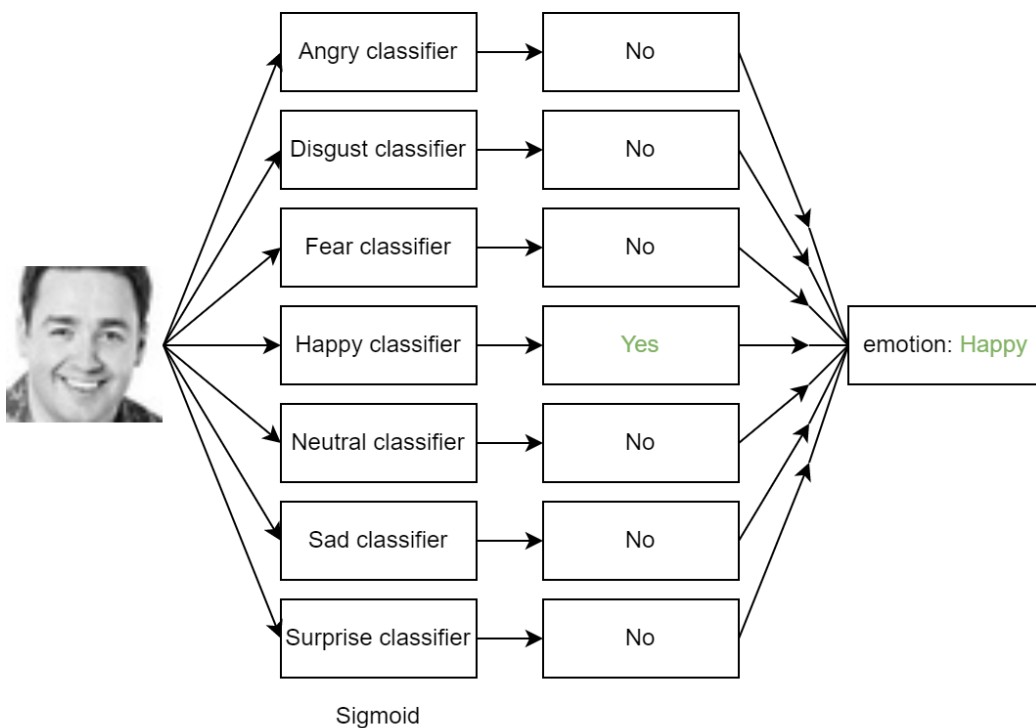

**Figure 5.** An overview of the simplified OVR architecture.

- Loss function was changed to binary cross-entropy;
- Output layer activation function was changed to Sigmoid, instead of the previously used Softmax;
- Transfer Learning models were also trained on 100 epochs, but for the binary implementations of five-layer and six-layer models, the number of epochs was increased to 200 each.

Binary models were built for three networks (five-layer, six-layer, and RESNET50) for each of the seven emotions. In total, 21 separate models were prepared and every one of them was trained on the corresponding binary dataset.

*4.5. Ensemble Models*

Ensemble models are a combination of multiple networks, as presenten on Figure 6. Each was trained separately and achieved satisfactory results. To further improve the overall performance, the model ensembling method was introduced, which consists of unweighted averaging of predictions from all ensemble members and then evaluating the final result. In this research, various combinations of such model ensembling are presented. During all experiments, the five-layer and six-layer models were combined only with each other. The same principle was applied for the transfer learning models. This method was used for both the original, filtered, and balanced dataset versions.

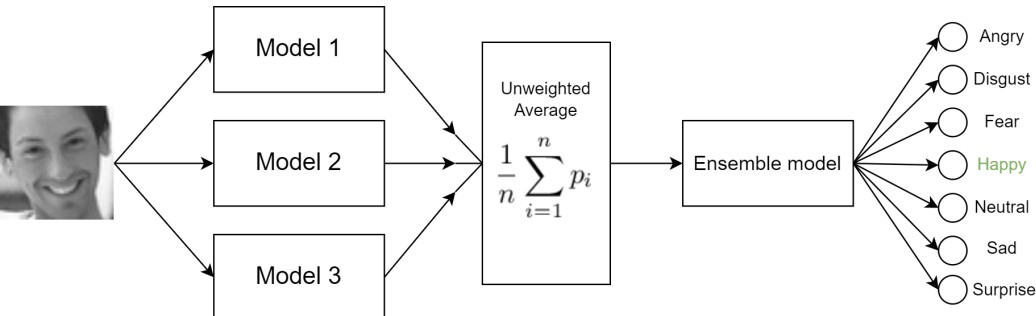

**Figure 6.** An overview of a simplified ensemble model architecture.

During the training of all models, the method of reducing the learning rate (LR) on the plateau was used.The validation accuracy was selected as the monitored quantity. In the case of learning stagnation (no model performance improvement for 5 epochs), the LR was decreased by a factor of 0.4. For the balanced datasets of the five-layer and six-layer models, both the original and filtered, the learning rate was reduced after 10 epochs of stagnation. While training, the method of creating checkpoints was also used for capturing the model with the highest validation accuracy. For each training, automatic Keras augmentation was used in the same manner as in the balanced datasets (in the case of balanced datasets, the Keras augmentation took place after the previous augmentation).

## 5. Experiments and Evaluation

The proposed CNN models were trained on Google Colab with a Tesla T4 GPU, with the use of Keras [43] and TensorFlow [44] as the backend. In order to provide the best evaluation, certain metrics were taken into consideration. At first, to compare the obtained results with other research, the accuracy was measured, as this seems to be the most popular metric for this task. The following formula was used:

$$Accuracy = \frac{TP + TN}{TP + TN + FP + FN} \tag{1}$$

To measure the number of correctly predicted emotions for the given sample from all the samples with a given emotion, the recall metric was introduced. Then, the precision metric was applied, which calculates the percentage of individual emotions predicted correctly among all other samples with the same predicted emotion. Due to the presence of both balanced and imbalanced datasets, the F1-score was also considered as the harmonic mean of the previously mentioned precision and recall. The formulas for precision, recall, and F1-score are as follows:

$$Precision = \frac{TP}{TP + FP} \tag{2}$$

$$Recall = \frac{TP}{TP + FN} \tag{3}$$

$$F1 = \frac{2 * Precision * Recall}{Precision + Recall} = \frac{2 * TP}{2 * TP + FP + FN} \tag{4}$$

To measure the inference time for each model, a set of 1000 randomly selected test images was prepared. Images were served in batches of 32. For predictions, the default Keras method—*predict* was singularly invoked 50 times, and its execution times were collected. Finally, the average of these iterations was computed, giving the resulting mean inference time. In order to present the model performance for each particular emotion, a normalized confusion matrix [45] was used, which can show the numeric prediction error between classes (Figures 7 and 8).

Below, the performance of each configuration is presented. Each table consists of the results for the particular models trained and evaluated on a particular dataset. First, the results on the original dataset (both the imbalanced and balanced version) are shown (Tables 9–13). Then, the performance of models trained on the filtered FER2013 is presented (Tables 14–18). Finally, an evaluation of the binary models is shown. The obtained results of the ensemble models made from networks presented in each table are also featured in Tables 10, 12 and 13 for the original FER2013 and Tables 15, 17 and 18 for the filtered FER2013.

**Table 9.** The obtained results on the original FER2013.

|  | Accuracy | Precision | Recall | F1-Score | Inference Time [s] |
|---|---|---|---|---|---|
| 5-layer | 0.6769 | 0.7248 | 0.6185 | 0.6646 | 0.6408 |
| 6-layer | 0.6763 | 0.7216 | 0.6136 | 0.6603 | 0.7547 |
| RESNET50 | 0.7272 | 0.7321 | 0.7227 | 0.7273 | 3.8706 |
| VGG16 | 0.7022 | 0.7419 | 0.6530 | 0.6926 | 4.8951 |

**Table 10.** The obtained results of ensemble models based on the original FER2013.

|  | Accuracy | Precision | Recall | F1-Score | Inference Time [s] |
|---|---|---|---|---|---|
| 5-layer + 6-layer | 0.6885 | 0.7305 | 0.6161 | 0.6653 | 1.0837 |
| RESNET50 + VGG16 | 0.7353 | 0.7606 | 0.7119 | 0.7348 | 7.7931 |

**Table 11.** The obtained results on the balanced, original FER2013.

|  | Accuracy | Precision | Recall | F1-Score | Inference Time [s] |
|---|---|---|---|---|---|
| 5-layer | 0.6977 | 0.7231 | 0.6639 | 0.6911 | 0.6245 |
| 6-layer | 0.6902 | 0.7085 | 0.6610 | 0.6830 | 0.7521 |
| RESNET50 | 0.7283 | 0.7339 | 0.7282 | 0.7310 | 3.8749 |
| VGG16 | 0.6944 | 0.7356 | 0.6467 | 0.6864 | 4.9040 |

**Table 12.** The obtained results of ensemble models based on the balanced, original FER2013.

|  | Accuracy | Precision | Recall | F1-Score | Inference Time [s] |
|---|---|---|---|---|---|
| 5-layer + 6-layer | 0.7008 | 0.7384 | 0.6650 | 0.6984 | 1.0892 |
| RESNET50 + VGG16 | 0.7425 | 0.7678 | 0.7197 | 0.7425 | 7.8611 |

**Table 13.** The obtained results of the ensemble of all models based on the original FER2013.

|  | Accuracy | Precision | Recall | F1-Score | Inference Time [s] |
|---|---|---|---|---|---|
| 4 custom models ensemble | 0.7041 | 0.7592 | 0.6504 | 0.6976 | 1.8862 |
| 4 TL models ensemble | 0.7506 | 0.7837 | 0.7155 | 0.7470 | 14.6228 |

**Table 14.** The obtained results on the filtered FER2013.

|  | Accuracy | Precision | Recall | F1-Score | Inference Time [s] |
|---|---|---|---|---|---|
| 5-layer | 0.7009 | 0.7472 | 0.6404 | 0.6875 | 0.7841 |
| 6-layer | 0.7061 | 0.7318 | 0.6620 | 0.6940 | 1.1109 |
| RESNET50 | 0.7477 | 0.7508 | 0.7449 | 0.7478 | 3.9211 |
| VGG16 | 0.7254 | 0.7579 | 0.6908 | 0.7218 | 4.8313 |

**Table 15.** The obtained results of ensemble models based on the filtered FER2013.

|  | Accuracy | Precision | Recall | F1-Score | Inference Time [s] |
|---|---|---|---|---|---|
| 5-layer + 6-layer | 0.7117 | 0.7537 | 0.6527 | 0.6975 | 1.5949 |
| RESNET50 + VGG16 | 0.7591 | 0.7775 | 0.7434 | 0.7597 | 7.5938 |

**Table 16.** The obtained results on the balanced, filtered FER2013.

|  | Accuracy | Precision | Recall | F1-Score | Inference Time [s] |
|---|---|---|---|---|---|
| 5-layer | 0.7143 | 0.7357 | 0.6875 | 0.7101 | 0.6266 |
| 6-layer | 0.7111 | 0.7331 | 0.6880 | 0.7092 | 0.7503 |
| RESNET50 | 0.7447 | 0.7482 | 0.7437 | 0.7459 | 3.9906 |
| VGG16 | 0.7178 | 0.7596 | 0.6710 | 0.7109 | 4.9960 |

**Table 17.** The obtained results of the ensemble models based on the balanced, filtered FER2013.

|  | Accuracy | Precision | Recall | F1-Score | Inference Time [s] |
|---|---|---|---|---|---|
| 5-layer + 6-layer | 0.7251 | 0.7491 | 0.6913 | 0.7180 | 1.1110 |
| RESNET50 + VGG16 | 0.7561 | 0.7727 | 0.7416 | 0.7565 | 7.8754 |

**Table 18.** The obtained results of the ensemble of all models based on the filtered FER2013.

|  | Accuracy | Precision | Recall | F1-Score | Inference Time [s] |
|---|---|---|---|---|---|
| 4 custom models ensemble | 0.7289 | 0.7625 | 0.6767 | 0.7154 | 2.2653 |
| 4 TL models ensemble | 0.7690 | 0.7970 | 0.7422 | 0.7678 | 14.6672 |

*5.1. Original FER2013*

As shown in Table 9, the highest accuracy achieved on the original imbalanced FER2013 by a single network was 72.72%, with the use of the RESNET50 (2.5% more than the VGG16). The 5-layer and 6-layer models performed similarly, with a slight advantage for the first, obtaining an accuracy around 67.66% ± 0.03%.

When it comes to the original balanced FER2013, the transfer learning models did not improve the performance (slightly better performance for the RESNET50 model and slightly worse for the VGG16 model). However, a significant performance improvement was noticed for the five-layer (+2.08%) and six-layer models (+1.39%). It is worth mentioning that the five-layer model trained on the balanced FER2013 was able to achieve a quite high result (69.77% accuracy) as a single network, and taking into account its considerably small architecture, the inference time of 0.6245s per 1000 test images is the lowest in the ranking.

As presented in Tables 12 and 13, the model ensembling method proved to be efficient in terms of achieving even higher accuracies but at a cost of inference time. Model ensembling was carried out in pairs - separately for the five- and six-layer models and transfer learning models. First, separate ensembles of five- and six-layer models and transfer learning models of the original imbalanced dataset were conducted. Then, the same approach was applied for the balanced original dataset. In general, the highest accuracy on the original FER2013 was achieved by an ensemble of four models (a pair of RESNET50 and VGG16 trained separately on balanced and imbalanced datasets). This approach achieved the best performance (75.06% accuracy) on the original FER2013 in this paper. Its confusion matrix is presented in Figure 7.

As shown in the confusion matrix in Figure 7, the "happy" and "surprise" emotions were the most recognizable. The highest misclassification error occurred between "angry" and "disgust", as well as "neutral" and "sad". The "fear" emotion was the least recognizable.

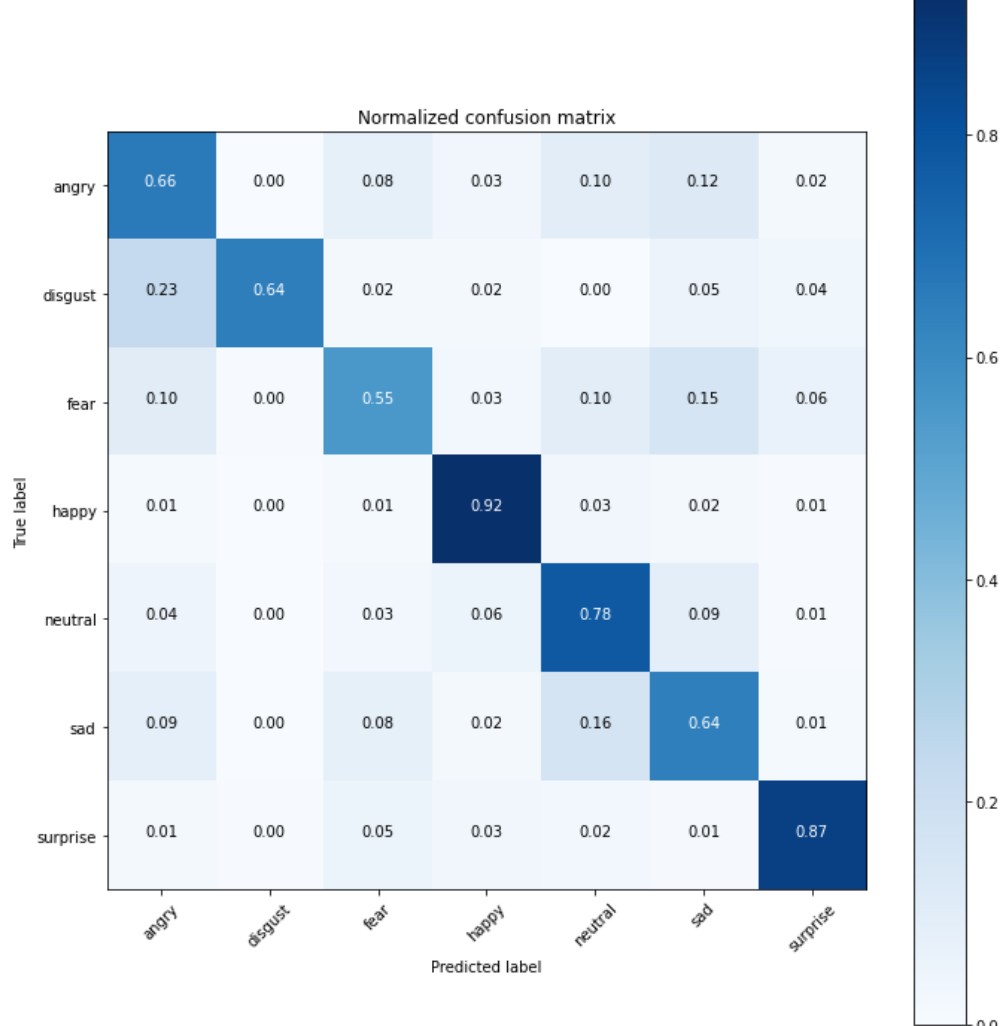

**Figure 7.** The confusion matrix of the transfer learning model ensemble (75.06% accuracy) for the original FER2013.

### 5.2. Filtered FER2013

As described in Section 3.3, the FER2013 was filtered by removing non-face images and subjectively relabeling clearly wrongly labeled images. This resulted in an improvement of the obtained performance. Considering the imbalanced dataset, 5-layer, 6-layer, RESNET50, and VGG16 gained 2.4%, 2.98%, 2.05%, and 2.32%, respectively.

In terms of the balanced dataset, a similar behavior as for its original version was noticed. A performance boost occurred for the five-layer and six-layer models, whereas for transfer learning models it was slightly worse. The highest accuracy achieved on this particular dataset was obtained in the same way as for the original one, by ensembling 4 transfer learning models, achieving 76.90%, which gives a +1.84% advantage over the original FER2013. A confusion matrix of this model is presented in Figure 8.

Compared to the most efficient model based on the original FER2013, a significant improvement was noticeable for the "angry" (±5%), "disgust" (±9%), and "sad" (±5%) emotions, whereas the rest of the emotions did not show any meaningful gain, while the inference time also did not change.

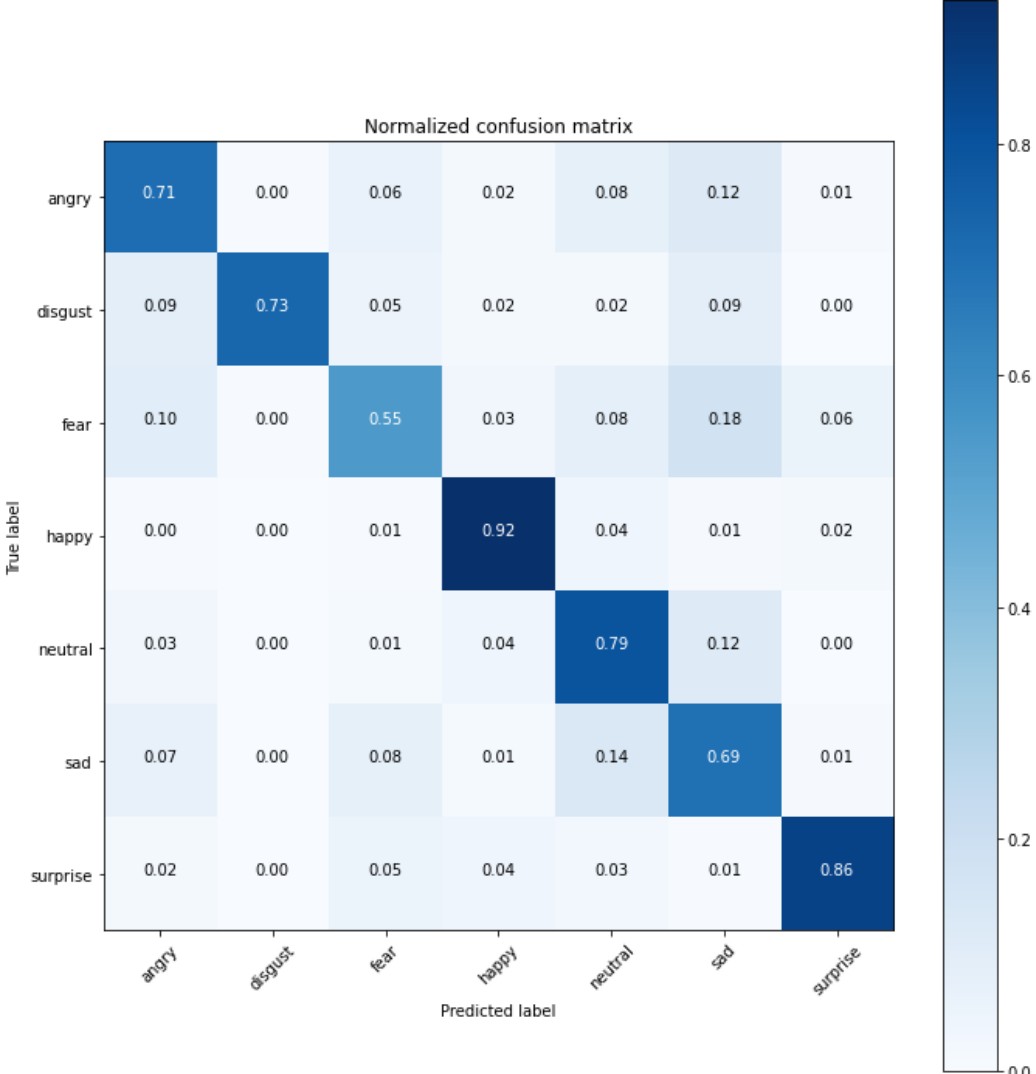

**Figure 8.** The confusion matrix of transfer learning models ensemble (76.90% accuracy) for the filtered FER2013.

### 5.3. Binary Datasets Based on the Filtered FER2013

Due to the results obtained in the previous tests, as well as the time and resource utilization, the VGG16 model was selected to be excluded from this experiment. The datasets used for this experiment were described in Section 3.5. Table 19 presents the accuracy obtained by each classifier for each base model.

**Table 19.** The obtained accuracies of the binary models for each base model.

|  | 5-Layer | Inference Time [s] | 6-Layer | Inference Time [s] | ResNet50 | Inference Time [s] |
|---|---|---|---|---|---|---|
| Angry Classifier | 0.7753 | 0.6513 | 0.7665 | 0.7690 | 0.8326 | 4.7803 |
| Disgust Classifier | 0.9018 | 0.6558 | 0.9018 | 0.7724 | 0.8750 | 4.7764 |
| Fear Classifier | 0.6946 | 0.6582 | 0.6731 | 0.7649 | 0.7774 | 4.7910 |
| Happy Classifier | 0.9173 | 0.6574 | 0.9249 | 0.7663 | 0.9343 | 4.7819 |
| Neutral Classifier | 0.8206 | 0.6547 | 0.8197 | 0.7686 | 0.8490 | 4.7758 |
| Sad Classifier | 0.7952 | 0.6610 | 0.7858 | 0.7763 | 0.8271 | 4.8075 |
| Surprise Classifier | 0.9072 | 0.6572 | 0.9059 | 0.7675 | 0.9227 | 4.7847 |

Considering only the two-class (positive and negative) classification problem, the binary models performed more accurately compared to their equivalents shown in the

confusion matrix in Figure 8. However, when the models were put together, in the sense of their results being concatenated without any voting, the performance decreased. Table 20 presents the obtained results.

**Table 20.** The obtained results of the ensemble of all binary models for each base model.

|  | Accuracy | Precision | Recall | F1-Score | Inference Time [s] |
|---|---|---|---|---|---|
| 5-layer | 0.6547 | 0.6196 | 0.6532 | 0.6238 | 0.9639 |
| 6-layer | 0.6591 | 0.6045 | 0.6484 | 0.6076 | 0.9863 |
| RESNET50 | 0.7278 | 0.7245 | 0.7160 | 0.7195 | 19.1138 |

Due to the fact that binary datasets were built based on the imbalanced, filtered FER2013, the comparison was conducted with results obtained on the same dataset and presented in Table 14. The 5-layer, 6-layer, and RESNET50 models showed a decrease of 4.62%, 4.70%, and 1.99%, respectively. Regarding the inference time, the five-layer and six-layer models were comparable with their equivalents. However, a huge difference was noticeable when it comes to RESNET50, whose inference time almost quadrupled. The ensemble of seven transfer learning models significantly increased the latency of the full model.

### 5.4. Comparison of Methods

Considering the fact that, in this research, the FER2013 was split randomly several times, the models were evaluated on different test sets. All came from the original FER2013 and were about the same size. A cross-validation technique was used, in order to achieve the most accurate results. The methods were compared based on the *accuracy* metric only, mainly due to the fact that the majority of other researchers did not apply additional metrics. The inference times for particular models were also not provided in the literature. Moreover, the inference time would have be difficult to obtain due to the lack of one, standardized computing platform among the researchers. The comparison of methods based on the original FER2013 is presented in Table 21.

**Table 21.** The comparison of methods based on the original FER2013.

| Method | Type | Accuracy | Auxiliary Data |
|---|---|---|---|
| Human Accuracy [5] | - | 65 ± 5% | × |
| "null model" [5] | Ensemble of 4 | 65.50% | × |
| Fard et al. [31] | RESNET50 | 68.25% | × |
| Kusuma et al. [22] | Single-model | 69.40% | × |
| Minaee et al. [27] | Attentional CNN | 70.02% | × |
| Porușniuc et al. [19] | RESNET50 | 71.25% | × |
| Jia et al. [24] | Ensemble of 3 + SVM | 71.27% | × |
| Fard et al. [31] | RESNET50 + Ad-Corre | 71.48% | × |
| Hua et al. [18] | Ensemble of 3 | 71.91% | × |
| Fard et al. [31] | Xception + Ad-Corre | 72.03% | × |
| Vulpe-Grigorași et al. [29] | Single-model | 72.16% | × |
| Kim et al. [15] | Ensemble of 36 | 72.72% | × |
| Nguyen et al. [11] | Single MLCNN | 73.03% | × |
| Khaireddin et al. [28] | Single-model | 73.28% | × |
| Connie et al. [16] | Ensemble of 3 | 73.40% | × |
| Jun et al. [17] | Single-model | 73.06% | × |
| Riaz et al. [23] | eXnet | 73.54% | × |
| Nguyen et al. [11] | Ensemble of MLCNNs | 74.09% | × |
| Khanzada et al. [25] | Ensemble of 7 | 74.80% | × |
| Pham et al. [26] | Single-model | 74.14% | ✓ |
| Pecoraro et al. [30] | Single-model | 74.42% | × |
| **Proposed Method** | **Ensemble of 4** | **75.06%** | × |
| Zhang et al. [13] | Multi-task model | 75.10% | ✓ |
| Pramerdorfer et al. [14] | Ensemble of 8 | 75.20% | × |
| Khanzada et al. [25] | Ensemble of 7 | 75.80% | ✓ |
| Pham et al. [26] | Ensemble of 6 | 76.82% | ✓ |

As shown in Table 21, the proposed method achieved a competitive accuracy of 75.06% as an ensemble of 4 models and without the use of extra training data.

Pramerdorfer et al. [14], similarly to this paper, did not apply any face registration and auxiliary data. However, the accuracy of 75.2% was obtained in their research as an ensemble of 8 models in total, which is twice as many as the proposed method. Considering the fact that their ensemble of eight models was composed of, among others, the VGG and RESNET networks, it seems highly likely that its inference time would be significantly higher than that of our method on comparable hardware. The two best models, which obtained accuracies of 76.82% and 75.8% were ensembles with a higher number of models and also utilized auxiliary training data, using additional FER datasets (as well as the model with the accuracy of 75.1%). This feature of the cited research made a comparison challenging. The latter, without the use of extra training data, achieved an accuracy of 74.8%, which is comparable with the proposed method and indicates possible future developments, in order to boost the prediction performance even higher.

Due to the fact that the proposed method utilized a lower number of models compared to the higher placed solutions with similar architectures presented in Table 21, it can be concluded that it is less complex and more time-efficient. Furthermore, as previously mentioned, it does not incorporate additional datasets, which is the main limitation of this study. The introduction of such supplementary training data constitutes a huge advantage over methods that do not make use of such. However, our main interest in this research was to utilize only the FER2013 dataset.

## 6. Discussion

Facial emotion recognition outside the laboratory environment has proven to be a quite challenging task. Different head poses, illuminations, and intensities of the expressed emotion, as well as samples presenting people of different ages, races, and genders, make the proposed solution more generalized and able to perform in real-life conditions.

We showcased the proposed methods, not only within the scope of the achieved performance in terms of correct predictions, but also within the scope of the obtained inference time, which can be crucial in modern real-time applications. This paper proposes a variety of approaches, including high-accuracy models with longer inference times and those whose classification performance is satisfactory but whose latency is more efficient.

The method of ensembling two models trained on an imbalanced dataset and the same two models trained on a balanced dataset resulted in achieving excellent accuracies of 75.06% and 76.90% on the original and filtered FER2013, respectively.

The fastest network in terms of inference time was the 5-layer model trained on the balanced original FER2013, obtaining a satisfactory accuracy of 69.77% as a single model (its equivalent trained on the balanced, filtered FER2013 achieved an accuracy of 71.43% with almost the same inference time).

In each case, the model ensembling technique improved the accuracy, but at a cost of latency. Dataset filtering increased the performance of each model even further. The final result of q 76.90% accuracy was higher than any presented in Table 21. The binary models performed exceptionally well as separate classifiers. However, the performance decreased after evaluation on the multi-class dataset (with concatenated results from each separate classifier). In future, this method will be further investigated to make the most of each particular binary model. For now, only basic result concatenation was utilized, without any voting.

## 7. Conclusions

This paper presented many approaches to the topic of facial expression recognition (FER) and utilized the FER2013 dataset, and FER2013 is one of the most widely used benchmark datasets. However, the original dataset suffers from many inconveniences, making its use challenging. In order to overcome these obstacles, various methods of data preprocessing were proposed. Starting with balancing datasets using data augmentation techniques, filtering the dataset to remove invalid or wrongly annotated images, or even creating binary datasets.

As for the models, four different types of CNN network were proposed, along with many combinations between them as ensembles. Model ensembling proved to be a highly performance-boosting technique, which made it possible to achieve an accuracy of 75.06% on the original FER2013 and 76.90% on the filtered FER2013. It is also worth mentioning that our proposed methods exceeded the capabilities of humans, who tended to correctly recognize emotion on the FER2013 dataset with a 65% accuracy.

The main limitation of this research is that it utilized only the FER2013 dataset. As can be seen in Table 21, the models obtaining higher accuracies than our solution mostly made use of additional training data coming from other datasets, which proved to be very beneficial for the resulting accuracy. What is more, this indicates that the proposed method would be able to achieve even higher performance by implementing such a strategy. However, in this paper, we focused only on making the best of the FER2013 dataset, without the use of extra training data.

This paper also took into consideration a binary classification, which has rarely been presented and discussed in existing studies. Our research proved that separate classifiers are worth considering, due to an increased accuracy compared to their equivalents in multi-class classification. Furthermore, an ensemble composed of such individual classifiers was examined. We did not observe any accuracy gain in this case; however, the use of more sophisticated voting might bring some performance benefits.

The prepared models obtain very good results in terms of prediction performance. Table 21 contains only the ultimate method presented in this paper; however, many other proposed solutions could also occupy a high place in this ranking. The custom CNNs proved to be capable of achieving accuracies around 70%, but at a much lower inference latency. Eventually, the final ensemble obtained an excellent result and was able to compete with the best models produced to date. Not only it is less complex, due to the utilization of a lower number of networks than other methods of similar architecture, but it also does not incorporate auxiliary training data. This proved that, besides the model choice (VGG16 and RESNET50 networks), the training strategy is also relevant. As opposed to other research, where several models were trained on the same dataset before ensembling, this paper presented a quite different approach. Each individual network was trained separately on differently preprocessed variants of the same dataset. More specifically, imbalanced and balanced versions of the FER2013 training set were prepared. It is worth noting that the test sets were uniform in both scenarios. As a result, each model branched into two independent ones, which were then further combined. Hence, the final ensemble did not consist of distinct networks but enabled the reuse of already prepared high-performing ones. Such a strategy allowed obtaining a favorable performance and utilizing a lower number of models in the final solution.

In this paper, a variety of results were obtained, presenting a wide range of achieved prediction accuracies, as well as the evaluation time of each model. Therefore, the trade-off between performance and inference latency was presented, allowing one to make more informed decisions on which model is best suited for a particular case. This allows a greater ability to properly apply the proposed methods to real-life scenarios.

**Author Contributions:** C.B. was responsible for the literature review, implementation, and analysis of results. He also wrote the relevant chapters. A.M. was responsible for substantive supervision, corrections, and coordination of the work. M.G. was responsible for scientific advice and proofreading. All authors have read and agreed to the published version of the manuscript.

**Data Availability Statement:** The research presented in this article was performed using the publicly available image dataset FER2013 [5]. All data modifications mentioned in the article have been made publicly available at: http://kt.agh.edu.pl/~matiolanski/FER2013_datasets/ (accessed on 1 May 2023).

**Acknowledgments:** This research was conducted as a bachelor thesis at the AGH University of Krakow. The publishing fees were kindly covered by www.aiseemo.com, Aissemo Sp. z o.o. The authors would like to thank these institutions for their organizational and financial support.

**Conflicts of Interest:** The authors declare no conflict of interest.

## Abbreviations

The following abbreviations are used in this manuscript:

CNN   Convolutional Neural Network
FER   Face Emotion Recognition
SGD   Stochastic Gradient Descent
SVM   Support Vector Machines

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
