# Peer review of "An Efficient Approach to Face Emotion Recognition with Convolutional Neural Networks"

_electronics, doi:10.3390/electronics12122707_

Round 1
Reviewer 1 Report
The manuscript entitled "An Efficient Approach to Face Emotion Recognition with Convolutional Neural Networks" aims to o investigate the possibility of further enhancement of Face Emotion Recognition solution in order to find an efficient approach in terms of both the achieved accuracy and the inference time by comparing the obtained results on several models, both a multi-class and a binary classification.
Although the material is well written and easy to follow, there are listed below some suggestions for the authors:
-in lines 74-139 the authors present some details about the accuracy and other evaluations. As they are very difficult to follow, I suggest authors to draw a table in which to summarize the information from the above mentioned lines.
-the authors should mention the name of the software used for data analysis
Major
-the literature review should be updated. For 41 sources, there is nothing from 2023, only 5 from 2022 and 6 from 2021. This topic is very actual and there is an abundant research. As I am allowed to suggest only 3 references, please search for others. My recommendations for inclusion, are: https://doi.org/10.3390/brainsci13040668 , https://doi.org/10.3390/info14010025 , https://doi.org/10.3390/electronics12051089
- what is the original contribution of this paper to the domain? the authors should specify , probably by the end of introduction.
- why did they used the FER2013 dataset although "suffers from many inconveniences making its use challenging"(lines 433-434) ? this selection should be explained and based on some valid criteria.
- the authors should elaborate more on the Conclusions. Also, they should specify the limitations of their study.
-general statements like this one "In conclusion, the goals of this research have been achieved." (lines 446-447) should be avoided and replaced with details on how the goal of the manuscript was achieved.
-Similarly, this following statement should be revised "Prepared models obtain satisfactory results in terms of prediction performance and inference time." (lines 447-448) by giving details about what are the "satisfactory results".
- Further, if the prepared models obtain JUST satisfactory results, why do we need this study? Good or excellent results should be obtained for this research to be published! please revise, I am sure you obtained very good results not just satisfactory. also, back up such statement by invoking a table with accuracy results of your model vs others.
Author Response
Please find our responses in a PDF file.

Reviewer 2 Report
The paper at hand proposes a solution for facial emotion recognition (FER) based on convolutional neural networks as well as the transfer and ensemble learning principles. More specifically, two custom CNN architectures along with the two well-established VGG and ResNet architectures are investigated for classifying 7 emotional classes on the FER2013 database. The above architectures are also combined in an ensemble classification setup and evaluated on the same database leading to the best-obtained results.
My main concern refers to the quite limited contribution of the presented work also reinforced by the poor presentation of the existing works in the field and the enhancements achieved by the introduced work.
To begin with, there is a high abundance of existing FER methods that exploit CNN architectures (specifically including the VGG and ResNet ones) to succeed high recognition performance on the FER2013. An ensemble of more CNN architectures is also extensively investigated. Considering the above, how this work differentiates from the existing literature?
In the last paragraphs of the introduction, the originality of the manuscript shall be displayed as well as the specific contributions (at least 2-3) of this work against the state-of-the-art have to be described.
In line 50 please provide references of the existing databases. For instance:
Mollahosseini, Ali, Behzad Hasani, and Mohammad H. Mahoor. "Affectnet: A database for facial expression, valence, and arousal computing in the wild." IEEE Transactions on Affective Computing 10.1 (2017): 18-31.
In lines 35-36: "Then the classification layers, based on the input data from previous ones, perform a classification (e.g. emotion recognition)." please cite:
Kansizoglou, Ioannis, et al. "Continuous Emotion Recognition for Long-Term Behavior Modeling through Recurrent Neural Networks." Technologies 10.3 (2022): 59.
In Table 21, we observe that the proposed work is not able to succeed the best results. Hence, what is the main benefit of this method in order to render it favorable against the existing ones? Is it less complex or computationally expensive? The authors could elaborate more on the above.
Additionally in Table 21, how do the authors explain the superiority of their method against existing ones using also an ensemble of similar architectures, e.g. [14]? Is there any innovation regarding the training strategy? The authors should clarify it more.
Personally, I am not able to perceive the usefulness of the custom 5-layer and 6-layer CNNs in the methodology section, given that they always lead to worse results. Practically, the presented work exploits two well-established models, i.e., the VGG and ResNet ones, fine-tunes them on the FER2013 database and combines them on an ensemble setup of classifiers. To my understanding, the custom CNNs constitute just an experimental study that empirically proves the superiority of the above pre-trained models, but they are not incorporated in the final optimal model.
Section 2 shall be rewritten in a more concise manner and organized in content-related paragraphs or subsections to enhance readability. In its current form, it just displays a list of existing works but no cohesion is observed among each work.
Please double-check the manuscript for typos.
Author Response

(The authors gave the same response as above.)

Round 2
Reviewer 1 Report
Based on the constructive review given by the reviewers, the authors have improved the manuscript. In this new version, I consider the article can be considered for being published within Electronics.
Author Response
Thank you for your positive review of our manuscript. We appreciate your kind words and support. Your feedback encourages us to continue our research and strive for excellence. We are grateful for your time and consideration.
Reviewer 2 Report
The revised manuscript is improved compared to the former version. My previous comments are addressed and the presentation is improved significantly, making the contributions of the paper clearer.
Minor Comment
The contributions in the last paragraphs of the Introduction could be read more easily in a bulleted list.
Please, unify Section 2 into fewer paragraphs based on the relation between the presented works.
Overall, I tend to accept this manuscript.
Please carefully double-check the manuscript for possible typos and syntax errors. For example,
Line 50: The citation number of JAFFE dataset is missing.
Line 222: Figure 2 (the first letter should be capitalized), according to the MDPI templates.
The same applies to Figure 5 in line 298. Please, check the entire manuscript.
Author Response
Dear Reviewer,
Thank you for your valuable feedback on our scientific paper. We appreciate your suggestions and have made the following revisions based on your comments:
-
Contributions: We have restructured the last part of the Introduction to present the contributions in a clear and concise bulleted list, making it easier to read and understand.
-
Section 2: We have reviewed the content in Section 2 and have condensed it into fewer paragraphs that are grouped based on the relationship between the presented works.
-
Proofreading: We have carefully double-checked the manuscript for typos and syntax errors. We have addressed the specific issues you pointed out, such as adding the missing citation number for the JAFFE dataset (Line 50) and ensuring the correct capitalization of Figure 2 (Line 222) and Figure 5 (Line 298) in accordance with the MDPI templates. Furthermore, we have thoroughly reviewed the entire manuscript for any other potential errors.